# Assessment of Cardiopulmonary Bypass Duration Improves Novel Biomarker Detection for Predicting Postoperative Acute Kidney Injury after Cardiovascular Surgery

**DOI:** 10.3390/jcm10132741

**Published:** 2021-06-22

**Authors:** Tao Han Lee, Cheng-Chia Lee, Jia-Jin Chen, Pei-Chun Fan, Yi-Ran Tu, Chieh-Li Yen, George Kuo, Shao-Wei Chen, Feng-Chun Tsai, Chih-Hsiang Chang

**Affiliations:** 1Kidney Research Center, Department of Nephrology, Chang Gung Memorial Hospital, Linkou Branch, Taoyuan 33305, Taiwan; kate0327@hotmail.com (T.H.L.); chia7181@gmail.com (C.-C.L.); Raymond110234@hotmail.com (J.-J.C.); franwis1023@gmail.com (P.-C.F.); yjtu1020@hotmail.com (Y.-R.T.); b9102087@yahoo.com.tw (C.-L.Y.); b92401107@gmail.com (G.K.); 2Graduate Institute of Clinical Medical Science, College of Medicine, Chang Gung University, Taoyuan 33302, Taiwan; 3Department of Cardiothoracic and Vascular Surgery, Chang Gung Memorial Hospital, Linkou Branch, Taoyuan 33305, Taiwan; josephchen0314@gmail.com (S.-W.C.); josephchen0939@yahoo.com.tw (F.-C.T.)

**Keywords:** liver-type fatty acid binding protein (L-FABP), acute kidney injury, biomarker, diagnostic performance

## Abstract

Urinary liver-type fatty acid binding protein (L-FABP) is a novel biomarker with promising performance in detecting kidney injury. Previous studies reported that L-FABP showed moderate discrimination in patients that underwent cardiac surgery, and other studies revealed that longer duration of cardiopulmonary bypass (CPB) was associated with a higher risk of postoperative acute kidney injury (AKI). This study aims to examine assessing CPB duration first, then examining L-FABP can improve the discriminatory ability of L-FABP in postoperative AKI. A total of 144 patients who received cardiovascular surgery were enrolled. Urinary L-FABP levels were examined at 4 to 6 and 16 to 18 h postoperatively. In the whole study population, the AUROC of urinary L-FABP in predicting postoperative AKI within 7 days was 0.720 at 16 to 18 h postoperatively. By assessing patients according to CPB duration, the urinary L-FABP at 16 to 18 h showed more favorable discriminating ability with AUROC of 0.742. Urinary L-FABP exhibited good performance in discriminating the onset of AKI within 7 days after cardiovascular surgery. Assessing postoperative risk of AKI through CPB duration first and then using urinary L-FABP examination can provide more accurate and satisfactory performance in predicting postoperative AKI.

## 1. Introduction

Acute kidney injury (AKI) is a common complication of hospitalization. The incidence of AKI in inpatients ranges from 5% to 10% and can reach as high as 50% among patients undergoing cardiac or aortic surgery [1,2,3]. AKI has a significant negative impact on patients’ short- and long-term outcomes, so the prevention and early diagnosis of AKI are crucial. Current global guidelines use the serum creatinine concentration and urine output as the definition and staging basis of AKI, but neither of these two markers is a sensitive or specific marker of AKI. Serum creatinine, not only filtered by glomerular tubules but also secreted by renal tubules, is influenced by multiple clinical variables, such as muscle mass, drugs, and dietary content, and is not significantly elevated until 48 h after a renal injury [4,5,6,7]. Because of delayed changes and low sensitivity and specificity of current markers, novel biomarkers have been investigated in the past few years to achieve more sensitive and earlier detection of AKI.

Liver-type fatty acid binding protein (L-FABP) is a member of the mammalian intracellular FABP family. The FABP family includes nine members distinguished according to tissue-specific distribution: L (liver), I (intestinal), H (muscle and heart), A (adipocyte), E (epidermal), IL (ileal), B (brain), M (myelin), and T (testis) [8,9]. Among these protein products, L-FABP is expressed not only in the liver but also in the intestine, pancreas, stomach, lung, and kidney. In these tissues, L-FABP plays a crucial role in regulating fatty acid metabolism and intracellular transport by binding fatty acids and transporting them to the mitochondria or peroxisomes; it is also involved in signal transduction pathways associated with the peroxisome proliferator-activated receptor [10,11]. Several recent studies have also revealed that L-FABP can be an effective endogenous antioxidant by binding to fatty acid peroxidation products and causing their excretion into urine [12,13]. In view of its characteristic of being secreted into the urine when the renal tubules are damaged, urinary L-FABP is considered a potential biomarker of AKI and has shown promising results in several clinical trials [14,15].

However, L-FABP examination has not been routinely used as a clinical tool for early AKI diagnosis because of its relatively high cost. Under this consideration, urinary L-FABP examination might be more cost-effective for use in the population with a high risk of AKI. Numerous clinical and operative factors influence postoperative AKI risk after cardiac surgery. Among these factors, several recent studies have concluded that longer duration of cardiopulmonary bypass (CPB) was associated with a higher postoperative renal failure rate [16,17,18]. The use of urinary L-FABP measurement in patients with longer CPB duration for prediction and early diagnosis of AKI seems to be a cost-effective choice. In this study, we first investigated the discriminative ability of urinary L-FABP for AKI prediction in patients receiving cardiovascular surgery and then examined the predictive performance of urinary L-FABP in patients who had longer CPB during cardiovascular surgery.

## 2. Materials and Methods

### 2.1. Data Source

This prospective study was performed in a cardiac surgery intensive care unit (ICU) of a tertiary care referral center in Taiwan between August 2015 and July 2018. The study protocol was approved by the Institutional Review Board of the Chang Gung Memorial Hospital (No.103-1993B). Patients who were admitted to the cardiac surgery ICU after cardiovascular surgery and provided informed consent were enrolled in this study. Patients who were aged <20 years old, had an estimated glomerular filtration rate (eGFR) less than 30 mL/min/1.73 m^2^, were receiving regular dialysis, had undergone previous organ transplant, or had confirmed AKI before cardiovascular surgery were excluded.

### 2.2. Data Collection and Definition

The demographic data, clinical characteristics, preoperative hemogram and biochemistry, surgery type, and surgery-associated factors were collected in a prospective database and evaluated retrospectively. The preoperative hemogram and biochemistry data were determined as the data to be measured within 1 week before surgery. If there were multiple available serum creatinine data during the week before cardiovascular surgery, the lowest creatinine level was defined as the preoperative creatinine level, and the eGFR was estimated on the basis of the defined value by using the Chronic Kidney Disease Epidemiology Collaboration (CKD-EPI) creatinine equation [19]. The left ventricular ejection fraction (LVEF) was measured by performing two-dimensional echocardiography before surgery. The age, creatinine, and ejection fraction (ACEF) scores were calculated as age (in years)/LVEF% + 1 (if creatinine > 2.0 mg/dL) [20,21].

### 2.3. Measurement of L-FABP Levels

To sequentially trace the changes in urinary L-FABP levels, fresh urine samples were collected in nonheparinized tubes via indwelling Foley catheters separately after cardiovascular surgery. The first samples were collected within 4 to 6 h postoperatively, and the second samples were collected within 16 to 18 h postoperatively. The collected urine samples were centrifuged at 5000× *g* for 30 min at 4 ℃ to remove cells and debris. The clarified supernatants were then stored at −80 ℃ before measurement. Urinary L-FABP levels were measured in duplicate by using commercially available enzyme-linked immunosorbent assay kits according to the manufacturer’s instructions (Norudia human L-FABP assay kit, Ibaraki, Japan). The standard level of urinary L-FABP to creatinine ratio is 8.4 μg/gCr or less, and the intra-assay coefficient of variation for urine L-FABP is ≤15%.

### 2.4. Outcome Definition

To determine the predictive value of urinary L-FABP for AKI, the primary outcome was defined as development of AKI within 7 days after cardiovascular surgery. Using the definition and staging criteria based on the Kidney Disease: Improving Global Outcomes (KDIGO) clinical practice guidelines for AKI, AKI was confirmed under either of the following conditions: serum creatinine level ≥0.3 mg/dL within 48 h; or 1.5-fold increase in serum creatinine from baseline within 7 days [22]. The urine output criteria of the KDIGO guidelines were not used in the present study. Other outcome measurements included in-hospital mortality, renal replacement therapy, length of ICU stay, and length of postoperative hospital stay. A previous study reported that CPB duration is a continuous variable of postoperative AKI [18]. To assess whether the predictive performance of urinary L-FABP is improved in patients receiving longer CPB, we used a CPB duration of 120 min as a cutoff value in the present study.

### 2.5. Statistical Analysis

We used Student’s t test to compare the normal distribution continuous variables between groups (i.e., patients with vs. without AKI), and used the Mann–Whitney U-test to compare the variables between groups incompatible to the normal distribution according to the Kolmogorov–Smirnov test. We used Fisher’s exact test to compare the categorical data between groups. The performance of L-FABP measured at different timepoints in discriminating AKI was assessed using the area under the receiver operating characteristic curve (AUROC). To determine the optimal cutoff value for urinary L-FABP in discriminating AKI, we estimated the Youden index, defined as (sensitivity + specificity) − 1. The optimal cutoff value for L-FABP corresponding to the Youden index achieves the maximum because it is the point that matches the highest sensitivity and specificity values at the same time. All statistical tests were two-tailed, and a *p* value of <0.05 was considered statistically significant. No adjustment for multiple testing (multiplicity) was used in this study. The statistical analyses were conducted using IBM SPSS Statistics 22 software (IBM, Armonk, NY, USA).

## 3. Results

### 3.1. Baseline Characteristics Analysis: Non-AKI vs. AKI Groups

A total of 144 consecutive patients receiving cardiovascular therapy were enrolled in the present study. The baseline characteristics, preoperative hemogram and biochemistry data, surgery type, and surgery-associated factors are summarized in Table 1. Among 144 enrolled patients, 59 (41.0%) had postoperative AKI. No significant differences in age, sex, or presence of diabetes mellitus were noted between the AKI and non-AKI groups. The preoperative hemogram data revealed that the patients with postoperative AKI had significantly lower preoperative hemoglobin (11.61 ± 2.35 g/dL in the AKI group vs.13.13 ± 2.16 g/dL in the non-AKI group; *p* < 0.001). No significant differences in preoperative white blood cell count, platelet count, or alanine aminotransferase level were noted between the two patient groups. Neither the preoperative serum creatinine level nor the preoperative eGFR converted by serum creatinine level via the CKD-EPI equation exhibited a significant difference between the patients with postoperative AKI and those without AKI. The preoperative LVEF exhibited no significant difference between patients with and without AKI.

The postoperative urinary-LFABP level and urinary L-FABP to creatinine ratio were significantly different between the AKI and non-AKI groups in the first and second timepoints.

Regarding the details of cardiovascular surgery, patients with postoperative AKI had significantly longer CPB times during cardiovascular surgery (165.80 ± 62.53 min in the AKI group vs. 118.88 ± 46.29 min in the non-AKI group; *p* < 0.001), and patients with AKI also had significantly longer clamp times than patients without AKI. In the present study, we observed that AKI group patients had significantly longer ICU stays compared to the non-AKI group, but there was no significant different in the mortality rate between the two groups. 

### 3.2. Characteristics of Patients Who Had CPB Durations Longer Than 120 min

A total of 85 patients had CPB durations longer than 120 min in the present study, including all of the patients receiving coronary bypass grafting and valve combination surgery and a majority of the patients receiving valve surgery. Among these 85 patients, the average CPB duration was 172.49 ± 49.22 min, and 46 patients had postoperative AKI, with an AKI incidence rate of 54.1%. The baseline characteristics, preoperative hemogram and biochemistry data, surgery type, and surgery-associated factors are summarized in Table 2. No significant differences in age, sex, presence of diabetes mellitus, and preoperative examination data were observed between patients with and without AKI. However, patients with postoperative AKI had a trend toward a lower preoperative hemoglobin level. While comparing the postoperative urinary L-FABP data, only the second timepoint postoperative urinary L-FABP and urinary L-FABP to creatinine ratio showed significant differences between AKI and non-AKI group patients. In the patients with CPB durations longer than 120 min, patients with AKI tended to have longer ICU stays.

### 3.3. Performance of L-FABP in Discriminating AKI

We examined the performance of urinary L-FABP for diagnosing AKI. Two urinary L-FABP samples were obtained at different timepoints. The first samples were collected 4 to 6 h postoperatively, and the second samples were collected 16 to 18 h postoperatively. In the whole study population, the AUROC of urinary L-FABP at the first timepoint was 0.598 (95% confidence interval [CI], 0.503–0.694), and the AUROC of urinary L-FABP at the second timepoint was 0.720 (95% CI, 0.633–0.807) (Figure 1a). Using urinary creatinine concentration at each timepoint as a quotient derived by dividing urinary L-FABP, we found that the urinary L-FABP/creatinine ratio showed better performance than the urinary L-FABP measurement. The AUROCs of the urinary L-FABP/creatinine ratio at the first and second timepoints were 0.627 (95% CI, 0.533–0.722) and 0.727 (95% CI, 0.643–0.811), respectively (Figure 1b).

We then examined the performance of urinary L-FABP in diagnosing AKI in patients with CPB duration longer than 120 min. The urinary L-FABP at the second time exhibited better performance in patients with CPB longer than 120 min—the AUROC of urinary L-FABP at the first timepoint was 0.579 (95% CI, 0.456–0.702), and the AUROC at the second timepoint was 0.742 (95% CI, 0.636–0.848; Figure 2a). The results of the urinary L-FABP/creatinine ratio at the second timepoint were also more favorable—the AUROCs of the urinary L-FABP/creatinine ratio at the first and second timepoints were 0.596 (95% CI, 0.475–0.718) and 0.751 (95% CI, 0.648–0.855), respectively (Figure 2b). The *p* values, sensitivities, specificities, and cutoff values are shown in Table 3.

## 4. Discussion

A total of 144 participants receiving cardiovascular surgery were enrolled in the present study. Three points are worth summarizing. First, the urinary L-FABP measured at 16 to 18 h after cardiovascular surgery was favorable for diagnosing postoperative AKI within 7 days. Second, the use of urinary creatinine to correct urinary L-FABP measurement postoperatively was more favorable for diagnosing AKI than the use of urinary L-FABP alone. Third, urinary L-FABP and urinary L-FABP/creatinine ratio at 16 to 18 h postoperatively exhibited favorable performance in patients with CPB durations longer than 120 min during cardiovascular surgery.

According to previous studies, L-FABP elevation can be detected immediately after surgery, and the median L-FABP peak level was detected approximately 6 h after surgery [23,24]. Although many studies have investigated the performance of urinary L-FABP in cardiac surgery, only a few of them have evaluated L-FABP 12 h or longer after surgery. In a meta-analysis, Ho et al. reported six studies validating the performance of L-FABP in the diagnosis or prediction of AKI onset within 24 to 72 h after cardiac surgery. Among these studies, four evaluated urinary L-FABP within 3 h after surgery, with AUROCs ranging from 0.52 to 0.85, and two trials examined L-FABP levels from 6 to 12 h after surgery, with AUROCs ranging from 0.66 to 0.76. The composite AUROC of urinary L-FABP for discriminating postoperative AKI in this meta-analysis was 0.72 (95% CI, 0.60–085) [25]. One study sequentially followed the urinary L-FABP level from immediately after cardiac surgery to 48 h later. The AUROCs of urinary L-FABP for predicting the onset of AKI within 48 h after cardiac surgery immediately after the operation and 3, 6, 18, 24, and 48 h postoperatively were 0.86, 0.85, 0.83, 0.76, 0.78, and 0.75, respectively [26]. Katagiri et al. had investigated urinary L-FABP in patients receiving cardiac surgery, and reported an AUC of 0.72 for L-FABP at 4 h after cardiovascular surgery and increased to 0.76 at 12 h. The results were similar to the present study [27]. Compared with the previous study focusing on predicting early-onset AKI (24 to 72 h postoperatively), our study investigated the ability of L-FABP to discriminate the AKI event within 7 days after cardiovascular surgery on the basis of KDIGO guidelines. By observing the urinary L-FABP data in the present studies, we found that both AKI and non-AKI group patients had elevated in urinary L-FABP within 4 to 6 h after cardiovascular surgery, but only the AKI group patients had persistent elevation of the urinary L-FABP after 16 h. This might be the reason that the urinary L-FABP sample obtained at 16 to 18 h exhibited more favorable performance than the one obtained at 4 to 6 h in the present study. The sample at 16 to 18 h exhibited favorable performance in discriminating postoperative AKI, with an AUROC of 0.720 (95% CI, 0.633–0.807). Overall, the performance of urinary L-FABP in discriminating AKI after cardiovascular surgery in the present study was comparable to that reported previously.

The fluctuating hemodynamic status and renal injury might influence the urine output amount, which sequentially varies the concentration of urinary biomarkers in patients receiving cardiovascular surgery. Previous research has suggested several approaches to reduce the impact of fluctuating urine volume on urinary biomarkers in a setting of rapidly changing eGFR, including calculation of an average biomarker excretion rate, estimation of total biomarker excretion over a certain time period, or normalization of a biomarker to urinary creatinine [28,29]. Taking these reports as a reference, we calculated the urinary L-FABP/creatinine ratio to minimize the influence of urine volume fluctuating on urinary biomarker concentration. By using urinary L-FABP/creatinine ratio, the performance of urinary L-FABP in discriminating postoperative AKI was further improved, with AUROCs of 0.627 (95% CI, 0.533–0.722) at the first timepoint and 0.727 (95% CI, 0.643–0.811) at the second timepoint.

Although identifying the postoperative AKI is crucial, the balance between cost and effective clinical decision making is also important. Considering the high cost of examining L-FABP, we attempted to distinguish the patients with a high risk of AKI, and examined whether urinary L-FABP was more accurate in those patients. Several studies have reported that a longer duration of CPB was independently associated with an increased likelihood of and more severe AKI [17,30,31]. Axtell et al. further pointed out that CPB duration is a continuous variable that can be used to predict postoperative AKI, with longer CPB duration during cardiac surgery associated with a higher rate of postoperative AKI. Another study by Paarmaan et al. concluded that the duration of CPB is a relevant factor for expression of biomarkers of renal tubular injury presumed to facilitate early detection of AKI after cardiac surgery [32]. A study comparing elective coronary revascularization with and without CPB reported more glomerular damage and tubular function damage in the CPB group based on the significant increases in microalbuminuria and free hemoglobin and changes in fractional excretion of sodium and free water clearance [33]. On the basis of this finding, Haase et al. proposed that longer CPB during cardiac surgery was associated with more abundant free hemoglobin release accompanied with free iron production. The free iron released from hemoglobin has the potential to catalyze the Haber–Weiss and Fenton reactions, whereby superoxide radical and hydrogen peroxide yield hydroxyl radicals and sequentially induce tubulotoxicity. Considering that the fatty acid binding proteins are not only endogenous antioxidants but also high-affinity heme-binding proteins, this renal tubular marker might be further associated with free iron–mediated toxicity, but not only to the tubular injury itself [34]. By combining this information, we evaluated the performance of urinary L-FABP and urinary L-FABP/creatinine ratio in patients with longer CPB duration during cardiovascular surgery and found that the AUROC at the second timepoint was further improved in patients with CPB longer than 120 min. According to this result, the use of urinary L-FABP examination for prediction and early diagnosis of postoperative AKI in patients with CPB durations longer than 120 min appears cost-effective.

A previous study revealed that preoperative risk assessment by ACEF score can improve urinary neutrophil gelatinase-associated lipocalin, another novel biomarker, in discriminating postoperative AKI in cardiac surgery patients [35,36]. However, in the present study, we used 1.1 as the cutoff ACEF score according to the previous report, the performance of urinary L-FABP could not be further improved through preoperative risk assessment with the ACEF score. The AUROCs in patients with ACEF scores ≥1.1 were 0.650 (95% CI, 0.504–0.796) at the first timepoint and 0.685 (95% CI, 0.543–0.827) at the second timepoint. The baseline characteristics stratified by ACEF score and the AUROCs for urinary L-FABP levels in discriminating AKI stratified by ACEF score are reported in Appendix A. According to this result, CPB time during cardiovascular surgery seems to be a more suitable risk assessment indicator than the preoperative ACEF score with clinically applied urinary L-FABP.

In the past few years, numerous biomarkers have been investigated to identify or predict AKI and each of them imply different injured segments [37]. Several studies had reported the comparison results between L-FABP and other AKI biomarkers. Katagiri et al. had examined the urinary L-FABP and N-acetyl-β-D-glucosaminidase (NAG) in cardiac surgery patients, which revealed that the L-FABP showed high sensitivity and NAG detected AKI with high specificity [27]. Schley et al. had reported that diagnostic performance 4 h after surgery of plasma NGAL, cystatin C, and L-FABP were AUROC 0.82, 0.76, and 0.73, respectively [38]. However, considering that these biomarkers are released from different segments and via various mechanisms during renal injury. For example, urinary L-FABP, kidney injury molecule-1 (KIM-1) from the proximal tubule, uromodulin (UMOD) is secreted from the loop of Henle and NGAL is released from the distal tubule. The importance of combining these biomarkers might not be to compare each of them but using the characteristics of each biomarker together to localize the specific segments of injured tubules and to figure out the pathophysiology process of AKI, according to the review published by Wen et al. [37]. In addition, more and more AKI biomarkers have been identified in the past few years. Several studies have proved that early biomarker-based prediction of AKI, followed by implementation of AKI management protocol, based on KDIGO guidelines, can improve AKI incidence, severity, length of ICU, and hospital stay [39,40,41]. For example, Rizo-Topete et al. set up a nephrology rapid response team (NRRT) which uses Nephrocheck to identify AKI risk and manage the moderate- and high-risk patients following KDIGO AKI guidelines. Compared to standard practice, the NRRT protocol has been proven to decrease the number of AKI patients and numbers of patients requiring renal replacement therapy [41]. In the present study, we had concluded that urinary L-FABP exhibited favorable performance in discriminating the onset of AKI within 7 days after cardiovascular surgery. To investigated the benefit of clinical implantation of L-FABP, further studies might be needed to investigate whether early detection and prediction of AKI by urinary L-FABP, combined with AKI management protocol, based on KDIGO guidelines, can improve the outcomes of patients after cardiovascular surgery.

This study has several limitations. First, the sample size is relatively small, and the etiologies of AKI might vary in different types of cardiac surgery. Thus, the applicability of our data to a specific type of cardiac surgery might be limited. Second, in this study, AKI was identified on the basis of KDIGO guidelines, but the urine output criteria were not included, which might underestimate AKI incidence. By using the urine volume information, urinary biomarkers can be calculated to average the biomarker excretion rate, which might further minimize the influence of urine volume fluctuation. Third, despite the fact that the time course of change depicted that L-FABP peaked at approximately 6 h after cardiac surgery, sequential measurement might further improve the discriminative ability. Fourth, our study focused on evaluating the performance of urinary L-FABP in a high-risk AKI population. Considering that the CPB duration is a continuous variable factor of postoperative AKI, CPB duration might influence the performance of urinary L-FABP in a continuous manner, which might be further evaluated in a future study.

## 5. Conclusions

The present study indicated that urinary L-FABP exhibited favorable performance in discriminating the onset of AKI within 7 days after cardiovascular surgery. Using the urinary L-FABP/creatinine ratio in spot urine samples can further improve the discriminatory ability of urinary L-FABP. The present study also revealed that using CPB duration can help identify cardiovascular surgery patients with a high risk of AKI. By assessing postoperative risk of AKI through CPB duration first and then using urinary L-FABP or urinary L-FABP/creatinine ratio for prediction and early diagnosis of AKI can provide more cost-effective and accurate risk discrimination than using urinary L-FABP examination alone. The present study not only revealed the favorable performance of urinary L-FABP in discriminating postoperative AKI diagnosis in a cardiovascular surgery population but also offered clinicians more information about choosing the appropriate population in which to use the urinary L-FABP examination, to balance clinical cost with effective decision-making. Further studies are required to investigate whether this strategy could improve outcomes of patients undergoing cardiovascular surgery by early prediction and diagnosis of postoperative AKI.

## Figures and Tables

**Figure 1 jcm-10-02741-f001:**
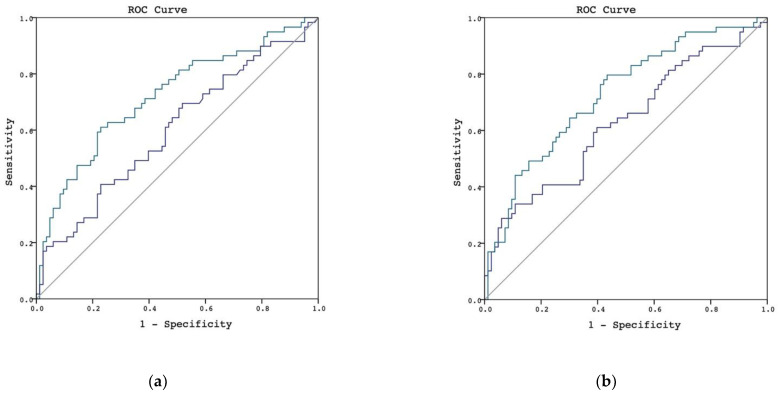
AUROC in discriminating postoperative AKI in whole patient population. (**a**) L-FABP in discriminating postoperative AKI, the urinary L-FABP at the first timepoint is shown by the blue line and the L-FABP at the second timepoint is shown by the green line. (**b**) L-FABP/creatinine ratio in discriminating postoperative AKI, the urinary L-FABP/creatinine ratio at the first timepoint is shown by the blue line and the L-FABP/creatinine ratio at the second timepoint is shown by the green line.

**Figure 2 jcm-10-02741-f002:**
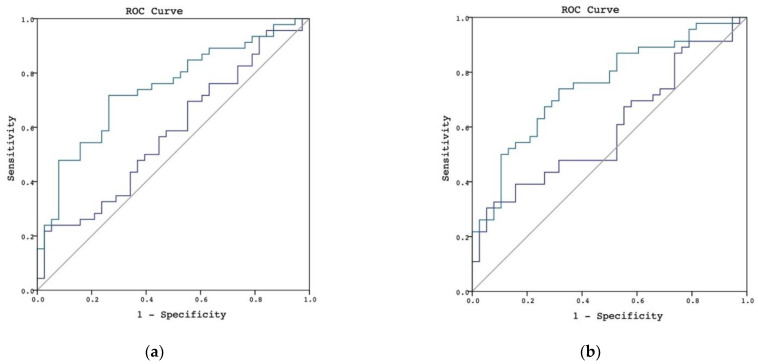
AUROC in discriminating postoperative AKI in patients with CPB durations longer than 120 min. (**a**) L-FABP in discriminating postoperative AKI in patients with CPB durations longer than 120 min, the urinary L-FABP at the first timepoint is shown by the blue line and the L-FABP at the second timepoint is shown by the green line. (**b**) L-FABP/creatinine ratio in discriminating postoperative AKI in patients with CPB durations longer than 120 min, the urinary L-FABP/creatinine ratio at the first timepoint is shown by the blue line and the L-FABP/creatinine ratio at the second timepoint is shown by the green line.

**Table 1 jcm-10-02741-t001:** Baseline characteristics and outcomes of the patients with AKI after cardiovascular surgery.

	All Patients (*n* = 144)	AKI (*n* = 59)	Non-AKI (*n* = 85)	*p*
Baseline characters				
Age, year	62.0 ± 12.8	63.6 ± 12.8	60.9 ± 12.8	0.216 ^‡^
Gender, male	95 (66.0%)	38 (64.4%)	57 (67.0%)	0.741
Underlying disease				
Diabetes mellitus	53 (36.8%)	20 (33.9%)	33 (38.8%)	0.547
Preoperation examination data				
Hemoglobin, g/dL	12.5 ± 2.4	11.6 ± 2.4	13.1 ± 2.2	<0.001 ^‡^
Platelet, 1000/uL	226.2 ± 81.8	223.1 ± 91.0	228.3 ± 75.2	0.715 ^‡^
White blood cell count, /uL	7443.4 ± 2599.3	7650.0 ± 2968.5	7302.3 ± 2322.2	0.434 ^‡^
Creatinine, mg/dL	0.8 [0.7; 1.0]	0.8 [0.7; 1.2]	0.8 [0.7; 1.0]	0.509 ^†^
eGFR, mL/min *	86.2 ± 22.7	83.0 ± 24.0	88.4 ± 21.5	0.159 ^‡^
ALT, mg/dL	25.0 [18.0; 36.5]	29.0 [18.0; 40.0]	24.5 [18.0; 35.2]	0.544 ^†^
LVEF, %	64 [52; 70]	62 [47; 68]	65 [55; 72]	0.211 ^†^
ACEF score	1.0 [0.8; 1.3]	1.1 [0.9; 1.3]	1.0 [0.8; 1.2]	0.091 ^†^
Surgical type				
Aortic surgery	1 (0.7%)	1 (1.7%)	0 (0%)	
CABG	53 (36.8%)	15 (25.4%)	38 (44.7%)	
CABG and valve surgery	9 (6.3%)	6 (10.2%)	3 (3.5%)	
Valve surgery	80 (55.6%)	36 (61.0%)	44 (51.8%)	
Others	1 (0.7%)	1 (1.7%)	0 (0%)	
Surgical-related factors				
CPB time, mins	138.2 ± 58.2	165.8 ± 62.5	118.9 ± 46.3	<0.001 ^‡^
Clamp time, mins	80 [0; 114]	98 [41; 143]	71 [0; 108]	0.003 ^†^
HTK perfusion	98 (68.1%)	46 (78.0%)	52 (61.2%)	0.034
Postoperative L-FABP data				
1st time L-FABP, ng/dL	59.8 [21.9; 179.0]	77.5 [26.7; 236.9]	43.0 [16.6; 124.1]	0.032 ^†^
2nd time L-FABP, ng/dL	60.4 [20.7; 165.2]	119.4 [50.2; 425.7]	42.6 [14.9; 97.9]	<0.001 ^†^
1st time L-FABP to Creatinine ratio, ng/mg	3.2 [0.9; 9.8]	5.2 [1.4; 13.5]	2.2 [0.7; 7.9]	0.006 ^†^
2nd time L-FABP to Creatinine ratio, ng/mg	0.8 [0.3; 2.8]	1.7 [0.7; 6.6]	0.5 [0.2; 1.4]	<0.001 ^†^
ICU stay, days	2 [1; 3]	3 [2; 5]	1 [1; 2]	<0.001 ^†^
Mortality	5 (3.5%)	1 (1.2%)	4 (6.8%)	0.072

Data were presented as frequency (percentage), mean ± standard deviation or median [25th, 75th percentile]. * pre-operation eGFR was estimated by creatinine via CKD-EPI equation; ^‡^ Student *t*-test; ^†^ Mann–Whitney *u*-test; ACEF score: age, creatinine, ejection fraction score; AKI: acute kidney injury; ALT: alanine aminotransferase; CABG: coronary artery bypass graft; CPB: cardiopulmonary bypass; eGFR: estimated glomerular filtration rate; HTK: histidine-ketoglutarate-tryptophan; ICU: intensive care unit; LVEF: left ventricular ejection fraction.

**Table 2 jcm-10-02741-t002:** Baseline characteristics and outcomes of the patients after cardiovascular surgery with bypass times over 120 min.

	All Patients (*n* = 85)	AKI (*n* = 46)	Non-AKI (*n* = 39)	*p*
Baseline characters				
Age, year	62.1 ± 12.1	63.5 ± 12.5	60.5 ± 11.5	0.269 ^‡^
Gender, male	55 (64.7%)	31 (67.4%)	24 (61.5%)	0.574
Underlying disease				
Diabetes mellitus	24 (28.2%)	14 (30.4%)	10 (25.6%)	0.625
Preoperation examination data				
Hemoglobin, g/dL	12.3 ± 2.3	11.7 ± 2.4	13.0 ± 2.0	0.013 ^‡^
Platelet, 1000/uL	220.8 ± 85.3	209.8 ± 83.9	233.8 ± 86.3	0.205 ^‡^
White blood cell count, /uL	7527.4 ± 2572.0	7880.0 ± 3201.8	7120.5 ± 2086.8	0.209 ^‡^
Creatinine, mg/dL	0.8 [0.7; 1.0]	0.8 [0.7; 1.2]	0.8 [0.7; 0.9]	0.412 ^†^
eGFR, mL/min *	85.5 ± 22.7	82.6 ± 23.1	88.9 ± 22.0	0.201 ^‡^
ALT, mg/dL	25.0 [16.0; 38.0]	31.0 [16.0; 40.0]	24.5 [16.8; 36.5]	0.508 ^†^
LVEF, %	60.9 ± 13.2	60.6 ± 13.9	61.2 ± 12.5	0.855 ^‡^
ACEF score	1.0 [0.8; 1.3]	1.0 [0.9; 1.3]	1.0 [0.8; 1.2]	0.173 ^†^
Surgical type				
Aortic surgery	1 (1.2%)	1 (2.2%)	0 (0%)	
CABG	15 (17.6%)	7 (15.2%)	8 (20.5%)	
CABG and valve surgery	9 (10.6%)	6 (13.0%)	3 (7.7%)	
Valve surgery	59 (69.4%)	31 (67.4%)	28 (71.8%)	
Others	1 (1.2%)	1 (2.2%)	0 (0%)	
Surgical related factors				
CPB time, mins	172.5 ± 49.2	184.3 ± 57.8	158.5 ± 32.0	0.012 ^‡^
Clamp time, mins	103.6 ± 54.0	109.5 ± 50.8	96.7 ± 47.7	0.278 ^‡^
HTK perfusion	74 (87.1%)	41 (89.1%)	33 (84.6%)	0.537
Postoperative L-FABP data				
1st time L-FABP, ng/dL	91.0 [26.1; 244.1]	114.1 [39.9; 283.2]	73.8 [22.8; 203.1]	0.169 ^†^
2nd time L-FABP, ng/dL	104.1 [36.9; 270.0]	155.2 [62.6; 467.7]	54.5 [22.6; 118.3]	<0.001 ^†^
1st time L-FABP to Creatinine ratio, ng/mg	5.8 [1.5; 12.3]	5.7 [1.6; 15.2]	5.8 [0.9; 10.2]	0.096 ^†^
2nd time L-FABP to Creatinine ratio, ng/mg	1.2 [0.5; 4.5]	2.7 [0.9; 8.6]	0.9 [0.3; 1.4]	<0.001 ^†^
ICU stay, days	2 [1; 4]	3 [2; 5]	1 [1; 2]	<0.001 ^†^
Mortality	3 (3.5%)	3 (6.5%)	0 (0%)	0.104

Data were presented as frequency (percentage), mean ± standard deviation or median [25th, 75th percentile]. * pre-operation eGFR was estimated by creatinine via CKD-EPI equation; ^‡^ Student *t*-test; ^†^ Mann–Whitney *u*-test; ACEF score: age, creatinine, ejection fraction score; AKI: acute kidney injury; ALT: alanine aminotransferase; CABG: coronary artery bypass graft; CPB: cardiopulmonary bypass; eGFR: estimated glomerular filtration rate; HTK: histidine-ketoglutarate-tryptophan; ICU: intensive care unit; LVEF: left ventricular ejection fraction.

**Table 3 jcm-10-02741-t003:** Urinary L-FABP and urinary L-FABP-to-creatinine ratio performance in discriminating AKI.

	Population	AUROC (95% CI)	*p* Value	Sensitivity (%)	Specificity (%)	Optimal Cut-Off ^†^
1st timepoint	Urinary L-FABP					
	Total	0.598 (0.503–0.694)	0.046	40.7	77.1	>132.34
	CPB duration > 120 min	0.579 (0.456–0.702)	0.063	21.7	97.4	>337.08
	Urinary L-FABP-to-creatinine ratio					
	Total	0.627 (0.533–0.722)	0.010	33.9	89.2	>11.842
	CPB duration > 120 min	0.596 (0.475–0.718)	0.131	39.1	84.2	>11.842
2nd timepoint	Urinary L-FABP					
	Total	0.720 (0.633–0.807)	<0.001	61.0	77.1	>101.77
	CPB duration > 120 min	0.742 (0.636–0.848)	<0.001	71.7	73.7	>101.77
	Urinary L-FABP-to-creatinine ratio					
	Total	0.727 (0.643–0.811)	<0.001	79.7	56.6	>0.612
	CPB duration > 120 min	0.751 (0.648–0.855)	<0.001	73.9	68.1	>1.063

^†^ According to the Youden index; CPB: cardiopulmonary bypass; L-FABP: Liver type fatty acid binding protein.

## Data Availability

Data available in a publicly accessible repository.

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
