# Peer review of "Assessment of Cardiopulmonary Bypass Duration Improves Novel Biomarker Detection for Predicting Postoperative Acute Kidney Injury after Cardiovascular Surgery"

_jcm, 2021, doi:10.3390/jcm10132741_

Round 1

Reviewer 1 Report

Very nice and interesting study. Can the Authors explain better how the prediction of AKI can modify the clinical management of these patients? What do they suggest for the high risk group to avoid the occurrence of AKI?

Author Response

Reviewer 1:

Very nice and interesting study. Can the Authors explain better how the prediction of AKI can modify the clinical management of these patients? What do they suggest for the high risk group to avoid the occurrence of AKI?    

Reply

Thank you for your kindly comment. According to the previous studies, the early prediction AKI combined with the KDIGO AKI care bundle can reduce AKI incidence, severity, ICU and hospital stays. For example, Ivan et al. had reported that using inhibitor of metalloproteinase-2 (TIMP2) and insulin-like growth factor–binding protein 7 (IGFBP7) as early biomarker to predict AKI in patients after major noncardiac surgery can reduced AKI severity, postoperative creatinine increase, length of ICU and hospital stays. [1] In San Bortolo Hospital in Vicenza, Rizo-Topete et al. had set up the nephrology rapid response team (NRRT) which using Nephrocheck (TIMP2 x IGFBP7) to early identified AKI risk and managed the moderate and high risk patients by the AKI management protocol based on KDIGO AKI guideline. Compared to the standard practice, NRRT protocol had decreased the number of AKI patients and numbers of patients requiring renal replacement therapy. [2,3] Based on these studies’ results, we expected that using L-FABP to early detection and prediction AKI after cardiovascular surgery combined with AKI management protocol based on KDIGO guideline might improving the prognosis in patient receiving cardiopulmonary surgery. Further studies will be needed to investigate the benefit of clinical implantation of L-FABP. For your insightful comment, we had added one short paragraph, the paragraph and modified sentence was in the page 9, line 320-332, in the revised manuscript as follows:

“Several studies have proved that early biomarker-based prediction of AKI followed by implementation of AKI management protocol based on KDIGO guideline can improve AKI incidence, severity, length of ICU and hospital stay. For example, Rizo-Topete et al. had set up the nephrology rapid response team (NRRT) which using Nephrocheck to identify AKI risk and manage the moderate and high risk patients following KDIGO AKI guideline. Compared to standard practice, the NRRT protocol has been proved to decrease the number of AKI patients and numbers of patients requiring renal replacement therapy. In the present study, we had concluded that urinary L-FABP exhibited favorable performance in discriminating the onset of AKI within 7 days after cardiovascular surgery. To investigated the benefit of clinical implantation of L-FABP, further studies might be needed to investigate whether early detection and prediction of AKI by urinary L-FABP combined with AKI management protocol based on KDIGO guideline can improve the outcomes of patients after cardiovascular surgery.”

Reviewer 2 Report

The authors examined the diagnostic performance of L-FABP for AKI after cardiovascular surgery. Although the novelty of the study itself is not that high, the focus on evaluation at a relatively late time after surgery, 16-18 hours, may be considered novel. However, the efficacy in actual clinical practice remains somewhat questionable.

  1. Katagiri et al. reported an AUC of 0.72 for L-FABP at 4 hours after cardiovascular surgery and an AUC of 0.76 at 12 hours (PMID 22269724). The expression of L-FABP is enhanced when the proximal tubule is subjected to ischemia or oxidative stress, and its urinary efflux is increased. From the results of this study, we can say that the timing is strongest at 16 hours?
  2. Has it been compared or combined with other urinary AKI biomarkers? If so, can you describe that as well?
  3. To what extent would an elevation of the biomarker at 16 hours be useful in clinical practice? For example, AKI occurring between 48 hours and 7 days after surgery is significantly more common than AKI occurring immediately after surgery, and is this clinically important? However, if most of the AKI can be detected by elevated sCre on the day after surgery, it may not be a strong message for a paper on urinary biomarkers.

Author Response

Reviewer 2:

The authors examined the diagnostic performance of L-FABP for AKI after cardiovascular surgery. Although the novelty of the study itself is not that high, the focus on evaluation at a relatively late time after surgery, 16-18 hours, may be considered novel. However, the efficacy in actual clinical practice remains somewhat questionable.

  1. Katagiri et al. reported an AUC of 0.72 for L-FABP at 4 hours after cardiovascular surgery and an AUC of 0.76 at 12 hours (PMID 22269724). The expression of L-FABP is enhanced when the proximal tubule is subjected to ischemia or oxidative stress, and its urinary efflux is increased. From the results of this study, we can say that the timing is strongest at 16 hours?

Reply

Thank you for your insightful comment. According to the previous studies, the median L-FABP peak level was detected approximately 6 hours after surgery. Only a few studies had series examined the urinary L-FABP after 12 hours or longer. And there was one study that followed urinary L-FABP level from immediately after cardiac surgery to 48 h later. And reported AUROCs at 3, 6, 18, 24, and 48 h postoperatively were 0.85, 0.83, 0.76, 0.78, and 0.75, respectively. [1] However, this study predicted AKI within 48 hours after surgery and our study tried to predict the AKI within 7 days following the KDIGO guideline criteria. As you kindly mentioned, Katagiri et al. had reported an AUC of 0.72 for L-FABP at 4 hours after cardiovascular surgery and an AUC of 0.76 at 12 hours, which is more similar to our result. [2] By observing the urinary L-FABP data in the present studies, we can find that both AKI and non-AKI group patients had elevated in urinary L-FABP within 4 to 6 hours after cardiovascular surgery, but the urinary L-FABP only keep elevated in AKI group patients after 16 hours. It might be possible that urinary L-FABP at 16 hours might be stronger while predicting postoperative AKI within 7 days. We had added this reference and modified some sentences, the modified sentence was in the page 7, line 237-239 and 243-246, in the revised manuscript as follows:

“Katagiri et al. had investigated urinary L-FABP in patients receiving cardiac surgery, and reported an AUC of 0.72 for L-FABP at 4 hours after cardiovascular surgery and increased to 0.76 at 12 hours.”

“By observing the urinary L-FABP data in the present studies, we found that both AKI and non-AKI group patients had elevated in urinary L-FABP within 4 to 6 hours after cardiovascular surgery, but only AKI group patients had persistent elevation of the urinary L-FABP after 16 hours.”

  1. Has it been compared or combined with other urinary AKI biomarkers? If so, can you describe that as well?

Reply

Thank you for your kindly comment. Several previous studies had compared the L-FABP with other biomarkers. Katagiri et al. had examined the urinary L-FABP and N-acetyl-β-D-glucosaminidase (NAG) in cardiac surgery patients, which revealed that the L-FABP showed high sensitivity and NAG detected AKI with high specificity. [2] Schley et al. had reported that diagnostic performance 4 hours after surgery of plasma NGAL, cystatin C, and L-FABP were AUROC 0.82, 0.76 and 0.73, respectively. But combinations of multiple biomarkers did not improve their diagnostic power. [3] According to the review published by Wen et al., different AKI biomarkers are released via various mechanisms during renal injury. For example, urinary L-FABP, kidney injury molecule-1 (KIM-1) from proximal tubule, uromodulin (UMOD) is secreted from the loop of Henle and NGAL is released from the distal tubule. Thus, the importance of combining these biomarkers might not to compare each of them but using the characteristics of each biomarker together to localize the specific segments of injured tubules and to figure out the pathophysiology process of AKI. [4] We added few sentences in revised manuscript to describe the relationship of these AKI biomarkers, the modified sentence was in the page 9, line 305-319, in the revised manuscript as follows:

“In the past few years, numerous biomarkers have been investigated to identify or predict AKI and each of them imply different injured segments. Several studies had reported the comparison results between L-FABP and other AKI biomarkers. Katagiri et al. had examined the urinary L-FABP and N-acetyl-β-D-glucosaminidase (NAG) in cardiac surgery patients, which revealed that the L-FABP showed high sensitivity and NAG detected AKI with high specificity. Schley et al. had reported that diagnostic performance 4 hours after surgery of plasma NGAL, cystatin C, and L-FABP were AUROC 0.82, 0.76 and 0.73, respectively. However, considering that these biomarkers are released from different segments and via various mechanisms during renal injury. For example, urinary L-FABP, kidney injury molecule-1 (KIM-1) from proximal tubule, uromodulin (UMOD) is secreted from the loop of Henle and NGAL is released from the distal tubule. The importance of combining these biomarkers might not to compare each of them but using the characteristics of each biomarker together to localize the specific segments of injured tubules and to figure out the pathophysiology process of AKI. according to the review published by Wen et al..”

  1. To what extent would an elevation of the biomarker at 16 hours be useful in clinical practice? For example, AKI occurring between 48 hours and 7 days after surgery is significantly more common than AKI occurring immediately after surgery, and is this clinically important? However, if most of the AKI can be detected by elevated sCre on the day after surgery, it may not be a strong message for a paper on urinary biomarkers.

Reply

        Thank you for your insightful comment. As you mentioned, serum creatinine is the most commonly used marker for AKI. However, creatinine exhibits no significant elevation until 48 hours after renal injury events, and its delayed elevations detrimentally affect the timely identification of renal injury. [5,6] According to previous studies and present study results, urinary L-FABP showed good performance on predicting AKI within 48 hours and within 7 days post operation. It makes early recognition and prediction of postoperative AKI become possible. Several studies have proved that early biomarker-based prediction of AKI followed by AKI management protocol based on KDIGO AKI guideline can improve AKI incidence, severity, length of ICU and hospital stay. [7-9] Based on these studies’ results, we hope the conclusion of present study, the positive result of diagnostic performance of L-FABP on postoperative AKI, might improving the prognosis in patient receiving cardiopulmonary surgery. Further studies might be needed to verify the benefit of clinical implementation of L-FABP combined with AKI management by KDIGO AKI guideline. For your insightful comment, we had added one short paragraph, the modified sentence was in the page 9, line 320-332, in the revised manuscript as follows:

“Several studies have proved that early biomarker-based prediction of AKI followed by implementation of AKI management protocol based on KDIGO guideline can improve AKI incidence, severity, length of ICU and hospital stay. For example, Rizo-Topete et al. had set up the nephrology rapid response team (NRRT) which using Nephrocheck to identify AKI risk and manage the moderate and high risk patients following KDIGO AKI guideline. Compared to standard practice, the NRRT protocol has been proved to decrease the number of AKI patients and numbers of patients requiring renal replacement therapy. In the present study, we had concluded that urinary L-FABP exhibited favorable performance in discriminating the onset of AKI within 7 days after cardiovascular surgery. To investigated the benefit of clinical implantation of L-FABP, further studies might be needed to investigate whether early detection and prediction of AKI by urinary L-FABP combined with AKI management protocol based on KDIGO guideline can improve the outcomes of patients after cardiovascular surgery.”

Reviewer 3 Report

Dear Authors,

The study was well designed and manuscript is clearly written. The authors have performed a careful study to examine whether urinary L-FABP examination can provide more accurate and satisfactory performance in predicting postoperative AKI.

I have only one suggestion to improve the manuscript:

Please add mean L-FABP and mean L-FABP to creatinine ratio values with units and normal range to Results section and table 1 and 2

Author Response

Reviewer 3:
The study was well designed and manuscript is clearly written. The authors have performed a careful study to examine whether urinary L-FABP examination can provide more accurate and satisfactory performance in predicting postoperative AKI.

I have only one suggestion to improve the manuscript:

Please add mean L-FABP and mean L-FABP to creatinine ratio values with units and normal range to Results section and table 1 and 2

Reply

Thank you for your kindly remind. We had added mean L-FABP and mean L-FABP to creatinine ratio values at first and second time point post cardiovascular surgery at Table 1 and 2, and we modified a few sentences of the result section to describe these findings. The standard level of L-FABP to creatinine ratio provided by the commercial kit was also added in our paragraph. The modified sentences were in the page 2, line 101-102; page 4, line 145-147 and page 5, line 169-171, in the revised manuscript as follows:

“The standard level of urinary L-FABP to creatinine ratio is 8.4μg/gCr or less and the intra-assay coefficient of variation for urine L-FABP ≤15%”

“The postoperative urinary-LFABP level and urinary L-FABP to creatinine ratio were significant different between AKI and non-AKI group in the second time point data, but there was no significant different in the first time point sample.”

“While comparing the postoperative urinary L-FABP data, there was only second time point postoperative urinary L-FABP showed significant different between AKI and non-AKI group patients.”
